

# Modelling take-off moment arms in an ornithocheiraean pterosaur

Benjamin W. Griffin[1,2], Elizabeth Martin-Silverstone[1], Rodrigo V. Pêgas[3], Erik Anthony Meilak[4], Fabiana R. Costa[3], Colin Palmer[1] and Emily J. Rayfield[1]

[1] Palaeobiology Group, School of Earth Sciences, University of Bristol, Bristol, United Kingdom
[2] School of Biological and Environmental Sciences, Liverpool John Moores University, Liverpool, United Kingdom
[3] Laboratory of Vertebrate Paleontology and Animal Behavior. Federal University of ABC, Alameda da Universidade, São Bernardo do Campo, SP, Brazil
[4] School of Pharmacy and Bioengineering, University of Keele, Keele, United Kingdom

## ABSTRACT

Take-off is a vital part of powered flight which likely constrains the size of birds, yet extinct pterosaurs are known to have reached far larger sizes. Three different hypothesised take-off motions (bipedal burst launching, bipedal countermotion launching, and quadrupedal launching) have been proposed as explanations for how pterosaurs became airborne and circumvented this proposed morphological limit. We have constructed a computational musculoskeletal model of a 5 m wingspan ornithocheiraean pterosaur, reconstructing thirty-four key muscles to estimate the muscle moment arms throughout the three hypothesised take-off motions. Range of motion constrained hypothetical kinematic sequences for bipedal and quadrupedal take-off motions were modelled after extant flying vertebrates. Across our simulations we did not find higher hindlimb moment arms for bipedal take-off motions or noticeably higher forelimb moment arms in the forelimb for quadrupedal take-off motions. Despite this, in all our models we found the muscles utilised in the quadrupedal take-off have the largest total launch applicable moment arms throughout the entire take-off sequences and for the take-off pose. This indicates the potential availability of higher leverage for a quadrupedal take-off than hypothesised bipedal motions in pterosaurs pending further examination of muscle forces.

## INTRODUCTION

'Portions of this text were previously published as part of a thesis ([https://research-information.bris.ac.uk/ws/files/348718716/Final_Copy_2022_12_06_Griffin_B_W_PhD_Redacted.pdf](https://research-information.bris.ac.uk/ws/files/348718716/Final_Copy_2022_12_06_Griffin_B_W_PhD_Redacted.pdf))'.

Powered flight is a method of locomotion that is limited to very few animals as it is energy intensive and requires specific adaptations to achieve launch, as well as deriving thrust and weight support *via* lift (*Rayner, 1989*). The most power-intensive part of powered flight is take-off from the ground. This stage requires the animal to get high enough into the air to utilise an unobstructed flapping cycle. Take-off also requires the animal to generate enough velocity such that the wings produce enough lift to overcome drag (*e.g.*, thrust)

Corresponding author
Benjamin W. Griffin,
b.w.griffin@ljmu.ac.uk

and support the weight of the animal (*Pennycuick, 1968*; *Rayner, 1988*; *Alexander, 1998*). The height and velocity requirements both increase with larger size, making take-off size-limiting for flying animals. No modern flying animal exceeds a mass of 25 kg with the heaviest volant living animal, *Otis tarda* (the Great Bustard), recorded as reaching 22 kg (*Henderson, 2010*). This limit has been previously attributed to the differential in scaling between increases in mass and increases in available muscle power which is predicted to increase at approximately mass$^{0.75}$ (*Alexander, 1998*). Despite this, many extinct animals have reached greater sizes and are still considered capable of flight, including birds such as *Argentavis magnificens* and *Pelagornis sandersi* which are predicted to have masses of 70 kg and 21.8–40 kg respectively (*Goto et al., 2022*). Pterosaurs vary in size, with medium sized pterosaurs reaching wingspans between 2 and 5 m predicted and masses ranging between 2 kg to 30 kg (*Witton, 2008*; *Martin-Silverstone, 2017*; *Goto et al., 2022*). Pterosaurs have also reached the largest sizes of any animal considered volant with the largest pterosaurs such as *Quetzalcoatlus northropi* predicted to have reached much greater masses (150 kg, or more commonly 250 kg (*Witton, 2008*; *Witton & Habib, 2010*; *Padian et al., 2021*)). Flight at such large body masses challenges our understanding of the functional limits of flight making understanding take-off in pterosaurs crucial to establishing the functional limits of flight in organisms.

There are two main hypotheses for how pterosaurs launched (*Habib, 2008*; *Witton, 2013*; *Padian et al., 2021*). The bipedal launch hypothesis is based on modern bird take-offs (*Padian, 1983*; *Earls, 2000*; *Chatterjee & Templin, 2012*; *Witton, 2013*; *Manzanera & Smith, 2015*; *Provini & Abourachid, 2018*; *Meilak et al., 2021b*; *Padian et al., 2021*) while the quadrupedal launch hypothesis is partially inspired by vampire bat terrestrial take-offs (*Schutt Jr et al., 1997*; *Habib, 2008*; *Molnar, 2009*; *Witton, 2013*; *Manzanera & Smith, 2015*; *Padian et al., 2021*). For an unassisted bipedal take-off, birds broadly fall into two different motions (*Earls, 2000*). In the first style, hereafter referred to as the bipedal countermotion take-off, the animal starts in a terrestrial locomotory bipedal pose. As the take-off cycle starts, the animal begins a crouching counter movement where it bends its hindlimbs and lowers its centre of mass while beginning to lean forward. The animal then rapidly extends the wings and hind limbs to launch, pushing the animal forward and upward. This take-off style is more favoured by birds that are less specialised for terrestrial locomotion, for example the European starling *Sturnis vulgaris* (*Earls, 2000*) or the magpie *Pica pica* (*Meilak et al., 2021a*). This style of take-off can also be seen from perches and has been examined in detail in the diamond dove *Geopelia cuneata* and the zebra finch *Taenopygia guttata* (*Provini & Abourachid, 2018*).

The second take-off motion is hereafter referred to as a bipedal burst take-off. This take-off begins already in a deep crouch and then rapidly extends the hind limbs with the body angled to launch nearly vertically while the wings start their initial downstroke. Because of the near vertical launch trajectory this take-off style results in limited forward motion but reaches greater heights. This style of take-off is favoured by birds that are specialised for living primarily terrestrially and fly rarely such as the European migratory quail *Coturnix coturnix* (*Earls, 2000*) or employed for rapid escape take-off as was examined in corvids (*Jackson & Dial, 2011*). A proposed mode of take-off was recently proposed for

the largest pterosaurs (*Padian et al., 2021*) which is nearly identical to the bipedal burst take-off. The only substantial difference between the bipedal burst take-off of birds and proposed pterosaur take-off is that pterosaurs could not start to utilise the wings to assist with the take-off until a sufficient height is reached for the wings to clear the ground. The distal wings of pterosaurs were unable to deform in the same manner as the feathers of a bird due to the bony spar that supports the pterosaur wing membrane so any contact with the ground during flapping would have likely damaged the wing (*Hone, Van Rooijen & Habib, 2015*). As this is the only distinction, we consider this model of pterosaur take-off as a bipedal burst take-off in our analysis.

The quadrupedal launch hypothesis described for pterosaurs is split into three main steps starting from a quadrupedal stance (*Habib, 2008*; *Molnar, 2009*; *Griffin et al., 2022*). The first is a crouching counter movement much like the bipedal take-off. When the deepest part of the crouch was reached the pterosaur began extending its hindlimbs providing an initial forward impulse and pushing the pterosaur onto its forelimbs. When the hindlimbs leave the ground the vault phase began. During this phase, the hindlimbs assumed the pose utilised in flight and the weight of the animal shifted to be entirely supported by the forelimbs. The launch phase then started as the forelimbs began to extend, pushing the pterosaur upwards and forwards until the forelimbs lost contact with the ground. This differs from the take-off utilised by most bats including the vampire bat *Desmodus rotundus* (*Schutt Jr et al., 1997*; *Riskin et al., 2006*; *Manzanera & Smith, 2015*) and New Zealand short-tailed bat *Mystacina tuberculata* take-off (*Hand et al., 2009*) where the take-off is almost vertical. In vampire bats the launch impulse is generated almost entirely by the forelimbs (*Schutt Jr et al., 1997*) instead of both the forelimbs and the hindlimbs as is hypothesised for pterosaurs (*Habib, 2008*; *Molnar, 2009*; *Witton, 2013*).

While the difference in the structural strength of pterosaur forelimb and hindlimb bones led to the original proposal of the quadrupedal take-off (*Habib, 2008*) and a recent study quantitatively investigated quadrupedal water take-off (*Pittman et al., 2022*), there has been very limited quantitative testing of the terrestrial take-off published (*Padian et al., 2021*; *Griffin et al., 2022*). Particularly of note, the ability of these take-off motions to generate the leverage that would be necessary to propel large pterosaurs into the air has not been quantitatively tested. One method for assessing leverage in extinct animals is the calculation of muscle moment arms (*Hutchinson et al., 2005*; *Bates et al., 2012*; *Maidment et al., 2014*; *Allen, Kilbourne & Hutchinson, 2021*; *Bishop, Cuff & Hutchinson, 2021*). While muscle lines of action have been presented previously for pterosaurs (*Fastnacht, 2005*; *Costa, Rocha-Barbosa & Kellner, 2014*) these studies focussed primarily on myological reconstruction for terrestrial locomotion and did not calculate the moment arms.

To test the ability of different pterosaur take-off hypotheses to produce leverage during the launch phase we have constructed the first OpenSim musculoskeletal model pterosaur. This model is based on a 5 m wingspan ornithocheiraean pterosaur. Using this model, we have estimated the take-off applicable muscle moment arms around each joint throughout the take-off motions for each of the hypothesised take-offs. Moment arm estimations serve as the first step in more complicated estimations of moment and power generation through take-off. However, we hypothesise that the leverage of the hindlimbs during the launch

phase would be greatest in the bipedal burst followed by the countermotion motions, as these motions rely on the leverage of the hindlimbs to power a bipedal take-off (*Meilak et al., 2021a*). Similarly, we treat the reverse scenario as a working hypothesis for the quadrupedal motion, that the leverage of the forelimbs will be greater in the quadrupedal launch phase than either of the bipedal take-offs as the forelimb motions are used in powering the take-off.

## MATERIALS & METHODS

### OpenSim modelling

An ornithocheiraean musculoskeletal model was constructed using μCT scans of SMNK-PAL 1133, an indeterminate ornithocheiraean pterosaur. The OpenSim model was based upon a skeletal model made in for a different study in 2015 (*Martin-Silverstone, 2017*; *Martin-Silverstone, Sykes & Naish, 2018*; *Griffin et al., 2022*). This specimen was selected due to the 3D preservation of the skeletal elements and relative completeness of the specimen. Further as a medium sized 5 m wingspan pterosaur the ability to successfully take-off can be more readily assumed than for the giant 10 m wingspan pterosaurs. The surface meshes from the skeletal model were checked and any errors were cleaned using (*3D Systems, 2014*) (3Dsystems, Morrisville, NC, USA). The articulated OpenSim model was constructed utilising the cleaned surface meshes and fitted geometric shapes following the workflow of *Meilak et al. (2021a)* using (*MATLABversion, 2021*; *Aherns, 2020*) (*Ahrens, Geveci & Law, 2004*), and OpenSim v4.0 and v4.1 (*Seth et al., 2018*). Due to the incomplete nature of SMNK-PAL 1133 scaled cylinders were added to represent the tibia and wing phalanges III and IV while duplicates of existing elements were used for missing vertebrae. The anterior skull is a upscaled version of AMNH FARB 24444 combined with the anterior section of SMNK 1133 (*Martin-Silverstone, 2017*; *Griffin et al., 2022*).

The joints of the articulated OpenSim model each include three rotational degrees of freedom (DOF). The individual DOF axes are aligned with specific movements of the limbs following the same definitions as *Kambic, Roberts & Gatesy (2014)*. Rotation along the abduction/adduction axis degree of freedom results in the limb moving away/towards the midline of the body. Flexion/extension axis rotation results in decreasing/increasing angles between the limbs. While long axis rotation along the axis rotates the distal element along the longest axis of the element. Not all DOFs included within the model are informative for take-off however all three DOFs in the shoulder and hip along with the flexion/extension DOF in the elbow, wrist, wing metacarpal, first wing phalanx, and knee are all expected to contribute to the take-off motion (Figs. 1 and 2).

### Muscle geometry

Twenty-two muscles related to forelimb motion and a further twelve muscles related to the movement of the hip and knee joints were modelled as muscle tendon units (MTUs) in the OpenSim model following the estimated lines of action between the origin and insertion points (Table 1 including abbreviations, Fig. 1). The MTUs were modelled based upon examination of physical specimens (Supplementary Material) and muscle reconstructions in the literature (*Dilkes, 1999*; *Bennett, 2001a*; *Bennett, 2001b*; *Bennett, 2003*; *Bennett, 2008*;

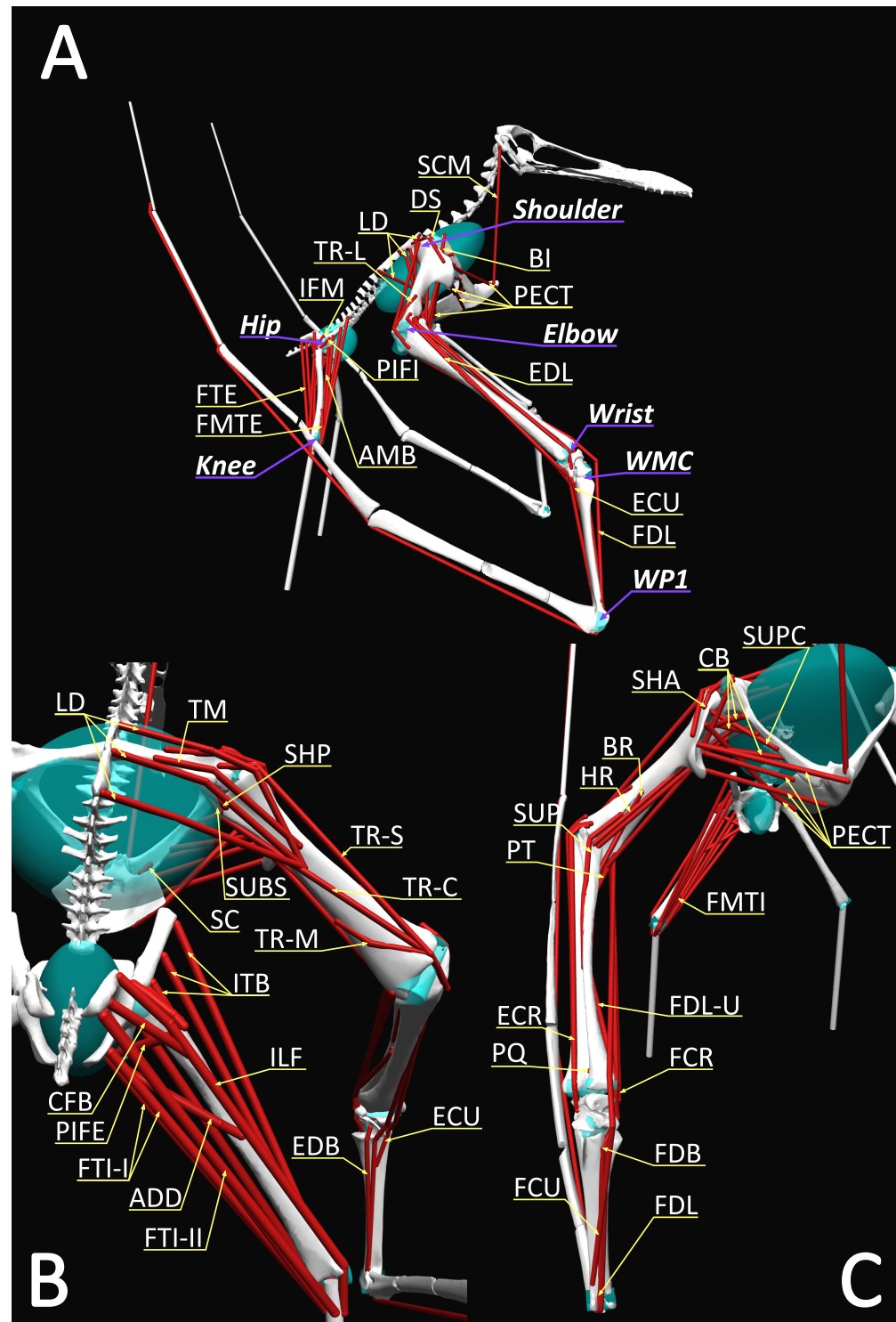

**Figure 1** Musculoskeletal model used in this study with labelled MTUs and joints in (A) lateral, (B) posterior and (C) anterior views. Muscle abbreviations follow the codes set forth in Table 1.

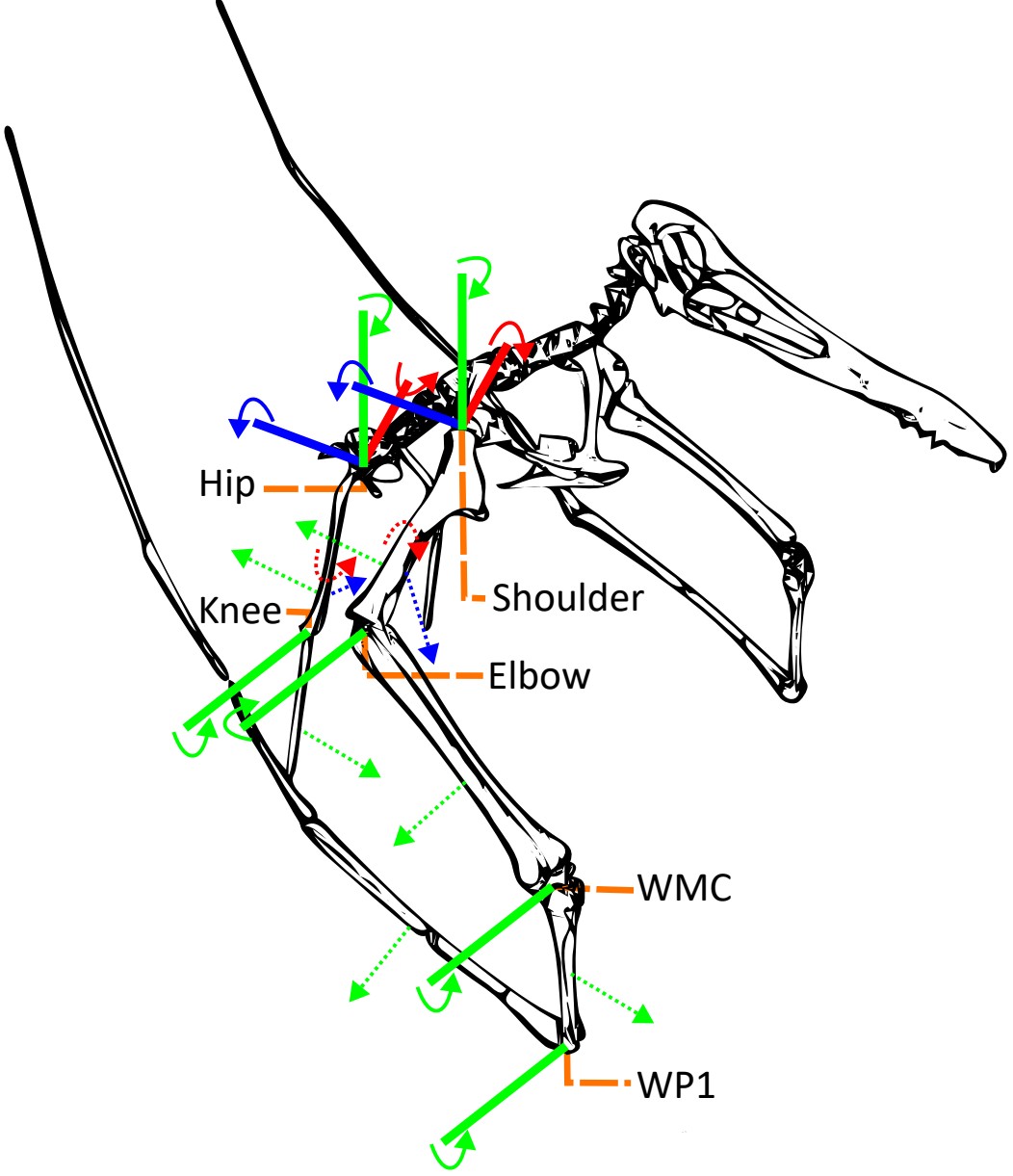

**Figure 2** **Simplified pterosaur model showing the directionality of the joint degrees of freedom examined in this study.** Solid arrows show the direction of positive rotation while the dashed arrows indicate the resultant movement of the element. Colours follow standard 3D space axes with positive $X$ axis rotation controlling long axis rotation (pronation, red), positive $Y$ axis rotation controlling extension (green), and positive $Z$ axis rotation controlling adduction (blue). A colour sensitive variant has been included in the supplementary material (Fig. S8).

*Fastnacht, 2005*; *Molnar, 2009*; *Witton, 2013*; *Costa, Rocha-Barbosa & Kellner, 2014*; *Tokita, 2015*; *Frigot, 2017*). For completeness, every muscle predicted to be involved in the take-off by these prior literature reconstructions (*Bennett, 2001a*; *Bennett, 2003*; *Bennett, 2008*; *Fastnacht, 2005*; *Molnar, 2009*; *Witton, 2013*; *Costa, Rocha-Barbosa & Kellner, 2014*; *Tokita,*

*2015*; *Frigot, 2017*) were included in our model. Inference levels for the presence of each muscle were determined following the extant phylogenetic bracket (EPB) inference model of *Witmer (1995)* and recorded in Table 1. Pterosaurs are bracketed by crocodiles and birds, following the most accepted interpretation of Pterosauromorpha as the sister-group of the Dinosauromorpha within Archosauria (*Ezcurra et al., 2020*; *Baron, 2021*; *Foffa et al., 2022*; *Kellner et al., 2022*). Muscles were only modelled when the inferred levels of confidence (as established by *Witmer 1995*) for their origin and insertion were assessed as either I or II (positive or equivocal assessment, respectively). Whenever direct correlates in the form of osteological markers could not be identified, apostrophes (as in I' and II') indicate that correlates were missing, but reconstruction is still carried out based on the myological patterns present in the phylogenetic bracketing groups. Table S1 summarizes all areas of origin and insertion, as well as their respective correlates (when present) and inference levels, for each reconstructed muscle. In the model, each origin and insertion point were placed at the centroids of the inferred areas of attachment with interpenetration between the bone meshes and the MTUs controlled by *via* points and wrapping surfaces (*Hutchinson et al., 2015*; *Modenese & Kohout, 2020*; *Bishop, Cuff & Hutchinson, 2021*; *Meilak et al., 2021a*; *Wiseman et al., 2021*).

To portray the complex lines of action more accurately in muscles with multiple origins such as the *m. triceps* and the *m. flexor tibialis internus* each muscle head was modelled individually (Table 1). For large muscles with broad attachment areas multiple lines of action were modelled at the cranial and caudal extents of the muscle in addition to a central muscle line of action. For the *m. pectoralis* lines of action were modelled at the cranial, caudal, medial, and lateral origin extent instead of a central muscle line of action, to better capture the broad origin and insertion of the muscle. The line of action moment arms were then averaged for these large muscles and this average value was used for examination of summed moment arms. In total, the present model includes 36 MTUs pertaining to wing musculature, and 18 MTUs related to the hip and knee musculature (Table 1).

## Kinematics

Key poses from the literature (*Bramwell & Whitfield, 1974*; *Padian, 1983*; *Fastnacht, 2005*; *Molnar, 2009*; *Chatterjee & Templin, 2012*; *Witton, 2013*; *Costa, Rocha-Barbosa & Kellner, 2014*; *Padian et al., 2021*) for the hypothesised take-offs were created for the OpenSim model and corrected to fit within the range of motion calculated for the model *via* a previous study using the ROM mapping methodology (*Griffin et al., 2022*). Intermediate poses were extrapolated using inverse kinematics in Maya and OpenSim to create a full kinematic profile of each take-off (Fig. 3). The timing between each pose was determined by relating the timing of each key pose of the model take-off sequences with the timing of the equivalent pose in the extant take-off sequences for the different take-off styles (*Schutt Jr et al., 1997*; *Earls, 2000*). The total time of the entire model sequence was then normalised as one second take-off motions. Directionality of joints are shown in Fig. 2.

The bipedal burst take-off timings are based on a quail profile (*Earls, 2000*) and the description by *Padian et al. (2021)*. The take-off has been split into three phases (Fig. 3A) starting with the crouch phase which begins in the fully crouched pose and lasts until

**Table 1** Modelled muscle tendon units (MTUs) for the OpenSim Ornithocheiraean model.

| Code | Muscle | Origin | Inference | Insertion | Inference |
|---|---|---|---|---|---|
| | | **Pectoral group** | | | |
| SCM* | sternocleidomastoideus | anterior sternum | I | squamosal | I' |
| SC* | sternocoracoideus | anterior margin of sternum | II' | coracoid | II |
| LD | latissimus dorsi | last cervical neural spine to distal notarium –3 MTUs | I' | dorsal (distal) humerus shaft scar | I |
| TM | teres major | posterolateral scapula | II | dorsal (proximal) humerus shaft scar | II |
| DS | deltoides scapularis | lateral scapula/acromion process | I | dorsal (anterior) deltopectoral crest | I' |
| SHA | scapulohumeralis anterior | scapula anterior to glenoid | II | dorsal (proximal) deltopectoral crest | II' |
| SHP | scapulohumeralis posterior | posterior margin scapula above glenoid | I | dorsal posterior process of humerus distal to SUBS | I |
| SUBS | subscapularis | medial ventral surface scapula | I | dorsal posterior process of humerus | I |
| TR-S | triceps | scapula - dorsal border of glenoid | I | olecranon process ulna | I |
| TR-C | triceps | coracoid - ventral posterior to glenoid | II' | olecranon process ulna | II |
| TR-M | triceps | medial - posterior side humeral shaft | I | olecranon process ulna | I |
| TR-L | triceps | lateral - anterior side humeral shaft | II | olecranon process ulna | II |
| PECT | pectoralis | sternum ventral –4 MTUs | I | entire ventral deltopectoral crest | I |
| SUPC | supracoracoideus | anterior ventral surface of coracoid | II | ventral proximal to deltopectoral crest | II' |
| CB | coracobrachialis | posterior ventral coracoid –3 MTUs | I | ventral posterior to deltopectoral crest | I |
| BI | biceps | coracoid biceps tubercule | I | proximal radius/ulna –2 MTUs | I |
| BR | brachialis | anterior humerus shaft | I' | proximal radius/ulna | I |
| HR | humeroradialis | proximal humerus distal to CB | II' | proximal radius | II' |
| FDL | flexor digitorum longus (quarti) | medial epicondyle of humerus and ulna | II | ventral extensor process WP1 and distal phalanges | I' |
| FDL-U | flexor digitorum longus (quarti) | medial ulna shaft | I | ventral extensor process WP1 and distal phalanges | I' |
| EDL | extensor digitorum longus (quarti) | lateral epicondyle of humerus | I | proximal posterior WP1 process | I' |
| FCU | flexor carpi ulnaris | medial epicondyle of humerus | I | anterior proximal WMC | II' |
| FCR | flexor carpi radialis | medial epicondyle of humerus | I | proximal anterior syncarpal | II |

**Table 1** (*continued*)

| Code | Muscle | Origin | Inference | Insertion | Inference |
|------|--------|--------|-----------|-----------|-----------|
| ECU | extensor carpi ulnaris | lateral epicondyle of humerus | I | posterior WMC large scar | II |
| ECR | extensor carpi radialis | lateral epicondyle of humerus | I | proximal posterior syncarpal | II' |
| SUP | supinator | ridge anterior to lateral epicondyle of humerus | I | $\frac{3}{4}$ length of posterior radius shaft | I |
| PT | pronator teres | medial epicondyle of humerus | I | posterior mid shaft | I |
| PQ | pronator quadratus | ulna shaft | I | posterior distal radius shaft | I |
| FDB | flexor digitorum brevis | dorsal distal syncarpal | II' | dorsal extensor process WP1 | I |
| EDB | extensor digitorum brevis | ventral distal syncarpal | II' | posterior WP1 process | II |
| | | **Pelvic Group** | | | |
| ADD | adductor femoris | lateral surface of the ischium | I | medial shaft (diaphysis) of the femur | I |
| IFM | illiofemoralis | lateral margin of preacetabular process of the ilium | I | greater trochanter | I |
| PIFE | puboischiofemoralis externus | lateral surface of the pubis | I | greater trochanter | I |
| PIFI | puboischiofemoralis internus | medial surface of ilium anterior to acetabulum | I | proximal surface of femur | I |
| AMB | ambiens | pubic tubercule | I | cnemial crest of tibia | I' |
| ITB | Iliotibialis | lateral margin of preacetabular process of the ilium −3 MTUs | I | cnemial crest of tibia | I' |
| FTE | flexor tibialis externus | lateral surface of the postacetabular process of the ilium | II | medial surface of tibia | II' |
| FTI-I | flexor tibialis internus | lateral surface of ischial tuberosity −2 MTUs | II | posteromedial shaft of tibia | II' |
| FTI-II | flexor tibialis internus 2 | lateral surface of the postacetabular process of the ilium | II | posteromedial shaft of tibia | II' |
| ILF | iliofibularis | lateral surface of the postacetabular process of the ilium | II' | posteromedial shaft of tibia | II' |
| CFB | caudofemoralis brevis | lateral iliac surface | I' | posterior (4th) trochanter of femur | I |
| FMTE | femorotibialis externus | proximal femoral shaft | I | cnemial crest of tibia | I' |
| FMTI | femorotibialis internus | proximal femoral shaft | I | cnemial crest of tibia | I' |

**Notes.**

*Indicates MTUs not directly related to the take-offs. Muscles using averaged MTUs include the number of MTUs used in the origin and insertion columns.
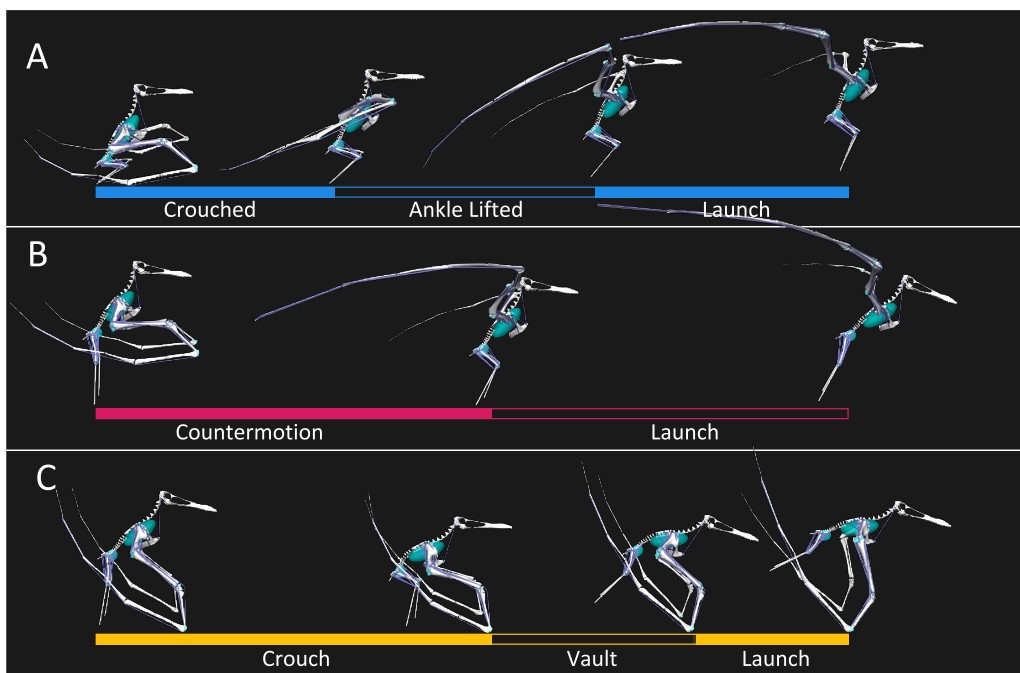

**Figure 3   One second take-off sequences used in this study highlighting key phases.** (A) Bipedal burst style take-off with crouched, ankle lifted, and launch phase timings highlighted. (B) Bipedal countermotion style take-off with countermotion and launch phase timings highlighted. (C) Quadrupedal take-off style with crouch, vault, and launch phases highlighted.

the ankle joint loses contact with the ground. From this point the second phase, termed the ankle lifted phase, contains the continued leg extension, finishing when the pterosaur reaches a fully digitigrade pose. The final phase is the launch phase where the leg extension moves the pterosaur from the digitigrade pose to the point where the feet lose contact with the ground and the wing moves to the start of the flight downstroke position.

The bipedal countermotion timings are based on a starling profile (*Earls, 2000*) and the earlier descriptions of pterosaur bipedal take-off poses (*Padian, 1983*; *Chatterjee & Templin, 2004*; *Chatterjee & Templin, 2012*; *Fastnacht, 2005*). The take-off has been split into two phases (Fig. 3B). The first phase is termed the countermotion phase and contains the starting bipedal stance through the flexion of the hindlimb and the unfurling of the wing. The second phase is the launch phase and includes the extension of the hindlimb and movement of the wing into the start of the flight downstroke position.

The quadrupedal take-off timing is based on a vampire bat profile (*Schutt Jr et al., 1997*) and primarily follows the description by *Habib (2008)* modified by other descriptions of key poses in the literature (*Fastnacht, 2005*; *Molnar, 2009*; *Witton, 2013*). The quadrupedal take-off has been split into three phases (Fig. 3C). The first phase is termed the crouch phase which begins in a quadrupedal stance pose and continues until the hindlimbs and forelimbs are fully flexed. The vault phase then includes the extension of the hindlimbs as the pterosaur pushes itself fully onto the forelimbs. The final phase is the launch, wherein the forelimbs extend until they leave the ground and the hindlimbs assume the pose that

will be utilised in flight. As the bat profile timing largely ignored the hindlimbs the timing of the hindlimb leaving the ground in the vault phase was added using the timing of the literature descriptions of the pterosaur quadrupedal take-off (*Habib, 2008*; *Molnar, 2009*) relative to the timing of this phase for the forelimbs.

### Moment arm analysis

Moment arms were recorded for each pose throughout the take-off kinematics in the shoulder, elbow, wrist, wing metacarpal, and wing phalanx 1 in the forelimb and the hip and the knee in the hindlimb. These joints were selected as they are the joints proposed to be utilised in the different launch hypotheses. The ankle joint was not included due to the lack of an accurate bone models of the tibia and metatarsals, preventing accurate mapping of the MTUs which adds a large amount of error to moment arm estimations (*Meilak et al., 2021a*). Moment arms were calculated and exported using the plotting tool in OpenSim for the kinematic sequence of each take-off in each joint degree of freedom (DOF). OpenSim calculates moment arms using the in-built virtual work methodology (*An et al., 1984*; *Pandy, 1999*; *Delp & Loan, 2000*; *Sherman, Seth & Delp, 2013*). Exported moment arms were then analysed using a modified R script (R version 4.1.2, Rstudio version 2021.09.2+382; *R Core Team, 2021*; *RStudio Team, 2021*) based upon the methodology of *Wiseman et al. (2021)*. As in the Wiseman methodology a Monte Carlo simulation of each muscle moment arm value was run wherein the value was independently allowed to uniformly randomise by values of up to $\pm20\%$ for 1,000 simulated trials to create error margins accounting for errors in moment arm estimation. The resultant distribution was then analysed for both the mean moment arm and the standard deviation. The mean moment arms for each muscle (Supplementary Material) were collated to determine the total summed moment arms. Summed moment arms were used as these show the likely directionality of the moment arm acting upon the bidirectional DOFs for the joints at each point in the launch hypothesis motions. This approach include the effects of the number of muscles acting upon the joint. Mean moment arms across muscles through each sequence which normalises the effect of muscle numbers are also included in the Supplementary Material.

## RESULTS

The following results apply the summed directional moment arms and associated estimation error calculated from the Monte Carlo approach plotted against the launch kinematics (Figs. 3–5). The entire kinematic sequence of each take-off was included despite the forelimbs being unable to be fully utilised in the bipedal take-off sequences and the hindlimbs no longer being in contact with the ground in the launch phase of the quadrupedal take-off sequence. These are included in the subsequent results figures despite not being considered launch applicable for completeness. Equally, though we consider only the moment arms in the hind limbs for the two hypothesised bipedal take-off kinematics and only the moment arms in the forelimbs following the onset of the vault phase in the hypothesised quadrupedal take-off kinematic sequence as launch applicable this does not mean these moment arms would all utilised while a pterosaur launched itself

into the air. Only the moment arm trends considered launch applicable are summarised below.

### Bipedal burst

The hip abductors through the bipedal burst kinematic initially decrease from the initial crouched pose (0.053 m) before plateauing in the ankle lifted phase of the take-off at 0.046 m. The adductors show a steady increase through the entire kinematic sequence from 0.105 m to 0.144 m. With regards to rotation the internal rotators of the hip decrease through the entire take-off sequence from 0.071 m to 0.046 m while the external rotators only begin to increase from 0.032 m noticeably during the ankle lifted phase to reach 0.058 m at take-off. The flexor moment arms in the hip slightly increase through the entire take-off kinematic from 0.054 m to 0.065 m as do the hip (0.098 m to 0.134 m) and knee extensors (0.046 m to 0.077 m). The knee flexors also sharply increase from 0.052 m to 0.084 m during the crouched phase before beginning to slow during the ankle lifted phase, eventually reaching 0.096 m.

### Bipedal countermotion

For the bipedal countermotion kinematic the moment arms of the hip abductors show an increase through the countermotion from 0.051 m to 0.060 m before decreasing in the launch phase to a final value of 0.052. The adductors show a slight decrease overall during the countermotion phase from 0.138 m to 0.130 m, increase slightly at the start of the launch phase to 0.139 m before ultimately decreasing further to 0.129 m. Regarding hip rotational moment arms the internal rotators are largely unchanged through the countermotion phase going from 0.048 m to 0.045 m and then increase slightly during the launch phase to reach 0053 m. The external rotator moment arms decrease throughout the countermotion phase from 0.130 m to 0.045 m and then increase through the launch phase to a value of 0.156 m. For the flexor moment arms there is a decrease from 0.081 m to 0.065 m through the countermotion phase and in increase through the launch phase reaching 0.087 m at take-off. This pattern is repeated in the hip extensor moment arms, decreasing from 0.134 m to 0.098 m before increasing until midway through the launch phase where the length of the moment arms peaks at 0.117 m and begins to decrease again to 0.111 m. The knee flexor moment arms increase through the countermotion phase from 0.041 m to 0.095 m and decrease through the launch phase to 0.038 m while the knee extensor moment arms remain largely unchanged, fluctuating between a low of 0.079 m and peak of 0.088 m throughout the take-off kinematic.

### Quadrupedal

Shoulder abductors through the quadrupedal take-off kinematic remain largely equivalent (varying from 0.028 m to 0.030 m) until the launch phase where they see a decrease in leverage with a value at take-off of 0.024 m. Conversely, the hip abductors increase until the hindlimbs leave the ground in the vault phase from 0.050 m to 0.060 m. The shoulder adductors show an increasing trend from 0.029 m to 0.031 m until the launch phase where it sharply decreases back to 0.023 m while the hip adductors decrease from 0.138 m until it reaches 0.129 m at the start of the vault phase. The shoulder internal rotation DOF

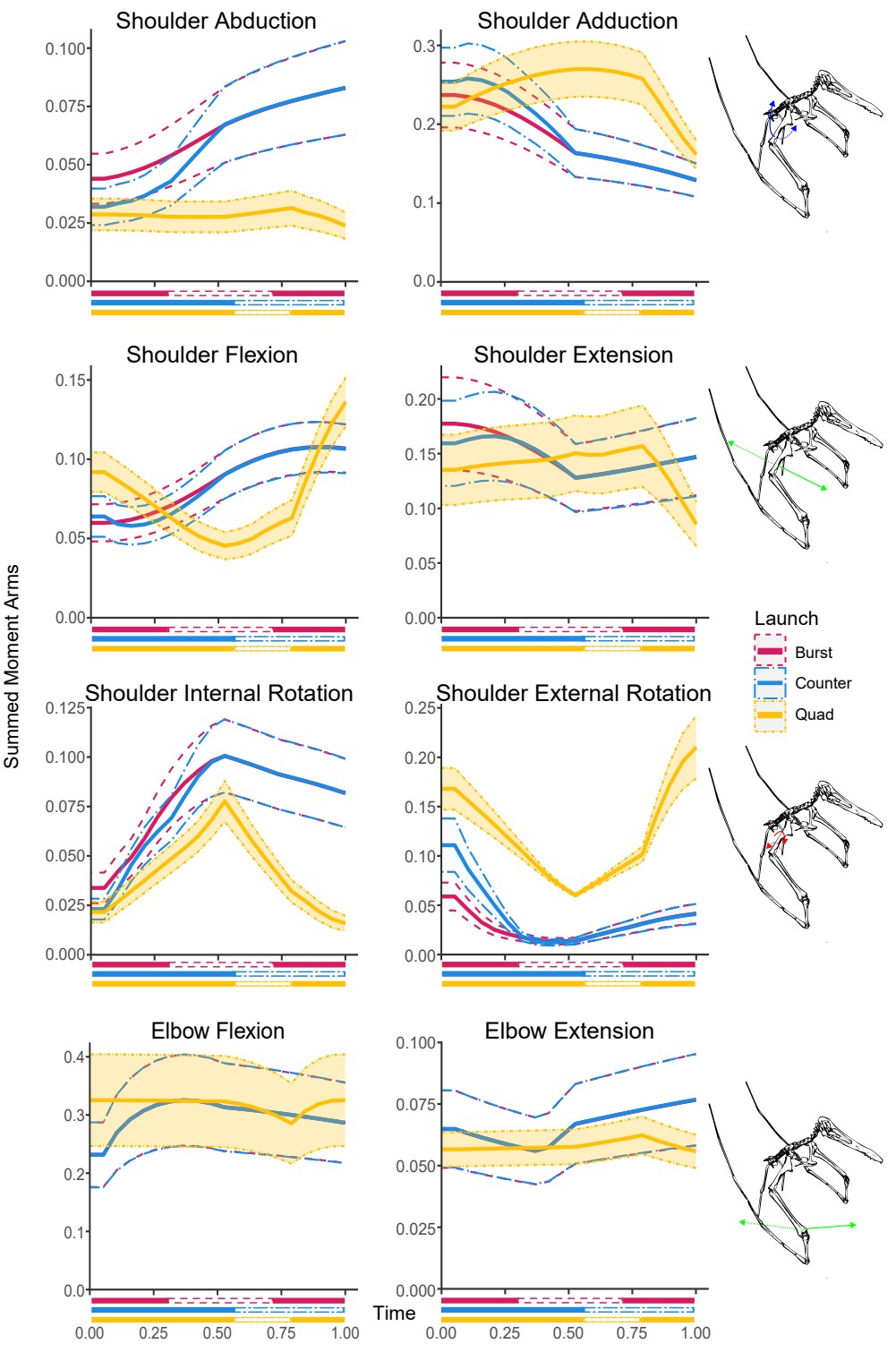

**Figure 4** **Summed moment arms in each hypothesised take-off motion for the shoulder and elbow rotational DOFs.** Solid lines indicate mean values following Monte Carlo simulation, dashed lines show estimated error, colouration indicates moment arm usage throughout the take-off. Take-off phase markers are equivalent to Fig. 3.

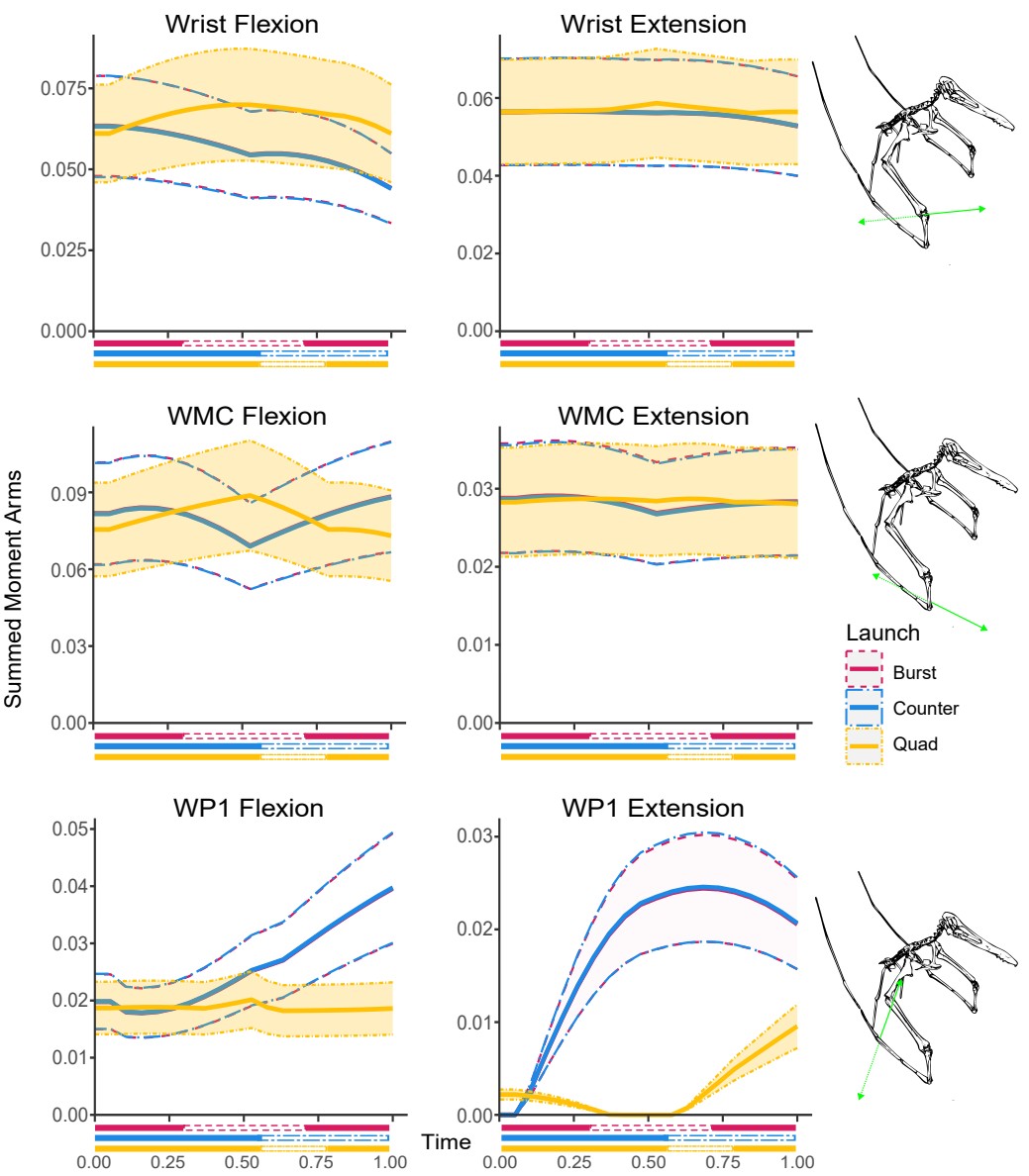

**Figure 5** **Summed moment arms in each hypothesised take-off motion for the lower forelimb rotational DOFs.** Solid lines indicate mean values following Monte Carlo simulation, dashed lines show estimated error, colouration indicates moment arm usage throughout the take-off. Take-off phase markers are equivalent to Fig. 3.

moment arms increase through the crouch phase from 0.022 m to 0.078 m and decrease through the rest of the take-off kinematic, reaching 0.016 m at take-off, while the external rotation DOF shows the reverse, decreasing from 0.168 m to 0.060 m then increasing to 0.210 m. The hip internal rotation feature a decrease from a start at 0.060 m to 0.054 m in the middle of the crouch phase but returning to a value of 0.056 m by the vault phase. The external rotation on the other hand features a pronounced decrease through the crouch phase going from 0.123 m to 0.045 m.

The shoulder flexion moment arms decrease through the crouch phase from 0.092 m to 0.045 m before increasing slowly in the vault to 0.063 m and rapidly in the launch phase to a final length of 0.136 m. The extensor moment arms slightly increase throughout the kinematic from a starting value of 0.135 m until the launch phase where they rapidly decrease from 0.157 m to 0.085 m. The elbow flexors slightly decrease through the crouch phase from 0.325 m to 0.323 m. This decrease becomes more pronounced in the vault phase reaching a low of 0.285 m before reversing in the launch phase to reach a final value of 0.325 m. The elbow extensor moment arms slightly increase throughout the crouch and vault phases from 0.057 m to 0.0623 m before decreasing during the launch phase to 0.056 m at take-off. The wrist flexion moment arms start at 0.061 m, increase during the crouch phase to 0.070 m and then decrease through the rest of the sequence to 0.061 m at take-off while the extensor moment arms are largely unchanged through the kinematic sequence varying between 0.056 m and 0.059 m. The wing metacarpal (WMC) moment arm trends are equivalent to the trends in the wrist with the flexion moment arms going from 0.076 m to a peak of 0.089 m and decreasing to 0.073, however the WMC extension moment arms are half a large as the equivalent wrist moment arms varying between 0.028 m and 0.029 m. The moment arms of the first wing phalanx (WP1) flexors are largely consistent through the crouch phase, varying between 0.019 m and 0.020 m before dipping slightly in the later phases of the take-off to reach a minimum value of 0.018 m. The extensors decrease until the end of the crouch phase from 0.002 m to 0 m and then increase again to a final length of 0.010 m. In the hindlimb, moment arms decrease in the hip flexors (0.75 m to 0.066 m) and extensors (0.144 m to 0.095 m) along with the knee extensors (0.086 m to 0.079 m) through the crouch phase while the knee flexors increase from 0.046 m to 0.094 m.

## DISCUSSION

The modelled launch motions include data for all of the muscles previous authors (*Bennett, 2001a*; *Bennett, 2003*; *Bennett, 2008*; *Fastnacht, 2005*; *Molnar, 2009*; *Witton, 2013*; *Costa, Rocha-Barbosa & Kellner, 2014*; *Tokita, 2015*; *Frigot, 2017*) have included in reconstructions of the fore and hind limbs. It must be noted however that despite including the moment arms of the forelimbs in the bipedal take-off models, the forelimbs would not be utilised at the point of take-off. In both bipedal take-offs the forelimbs are not used until the initial downstroke of the wings by which time launch has already occurred (*Padian, 1983*; *Padian et al., 2021*). Similarly, the quadrupedal take-off does not utilise the hindlimb moment arms at the point of launch. The quadrupedal take-off only utilises the hindlimbs until they lose contact with the ground during the vault phase of the take-off (*Habib, 2008*). This limitation on the availability of leverage generated by the launch underlies the hypothesis that the bipedal launches would generate a noticeable increase in hindlimb moment arms during the launch phase and the quadrupedal would have greater leverage in the forelimb. The results of our models (Figs. 4–6) do not find a large increase in the leverage of the hindlimb in the bipedal take-offs relative to the quadrupedal take-off. Equally, while the quadrupedal motion does have slightly larger moment arms in some DOFs it is not consistent. Despite this the results do show a noticeable difference between the total lengths of the moment arms acting on the fore and hindlimbs.

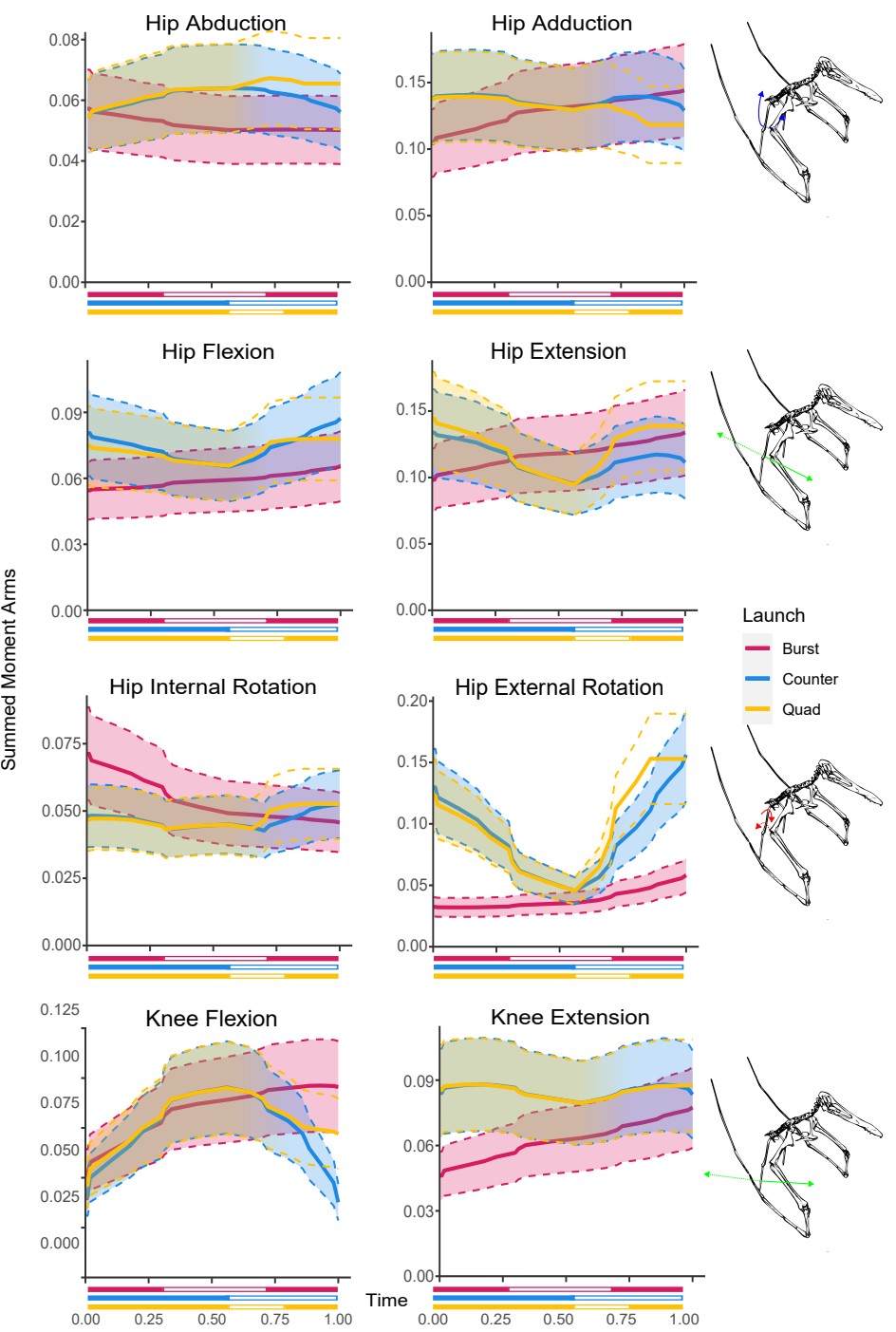

**Figure 6** **Summed moment arms in each hypothesised take-off motion for the hindlimb rotational DOFs.** Solid lines indicate mean values following Monte Carlo simulation, dashed lines show estimated error, colouration indicates moment arm usage throughout the take-off. Take-off phase markers are equivalent to Fig. 3.

The largest moment arms that occur for each of the simulated take-offs occurs in the upper forelimb joints, specifically in the elbow flexion/extension DOF and the shoulder

abduction/adduction DOF (Fig. 4). The largest moment arms across all the motions occur in the quadrupedal take-off profile (0.325 m in elbow flexion, and 0.270 m in shoulder adduction). The largest moment arms in the hindlimbs occur in each of the hip DOFs with 0.156 in external rotation for the countermotion take-off, 0.144 m extension in the quadrupedal take-off, 0.144 m adduction in the burst take-off motion (Fig. 6). The moment arms are largely equivalent between the different launches; except for the hip external rotation DOF which has a smaller moment arm during the burst take-off (Fig. 6). Overall, the largest hindlimb moment arm is half the length of the largest forelimb moment arm. Due to the likely inability of the bipedal take-off motions to fully access the larger moment arms of the forelimb without damaging its wings (*Hone, Van Rooijen & Habib, 2015*), the quadrupedal take-off would have access greater leverage simply because of the larger moment arms in the forelimb than the hindlimbs. The smallest moment arms occur in the extensional DOF of WP1 in the lower forelimb (Fig. 5). If only extensional moment arms are considered the combined forelimb moment arms in quadrupedal take-off remain slightly larger (0.235 m at the point of launch *vs* 0.195 m for the countermotion and 0.211 m for the burst take-off, see Supplemental Information). However, it should also be noted that for complex motions like jumping muscle function and moment generation are not confined to the sagittal plane so this might be an oversimplification, as was shown for take-off jumps of magpie (*Pica pica*) (*Meilak et al., 2021a*) and zebra finch (*Taeniopygia guttata*) (*Meilak et al., 2021b*).

Trends within the take-off moment arms closely match phase changes within the different take-off kinematics in all but the burst take-off hindlimb DOFs (Fig. 6). There is distinct overlap in the forelimb DOFs for the bipedal take-off motions which correspond with the wings gaining sufficient clearance to fully open without striking the ground. Similarly, both the bipedal countermotion and quadrupedal take-off hindlimb DOFs strongly overlap during the countermotion phases of each take-off kinematic before diverging during the later phases. The peak moment arms for the bipedal burst take-off tend to occur at launch or at the start of the take-off sequence. The peak moment arms of the bipedal countermotion take-off tend to occur during the countermotion phase or at launch except for the hip abduction and extension degrees of freedom where the peak occurs at the start of the sequence. The quadrupedal take-off peak moment arms occur at launch or the end of the crouch phase in the forelimb and at the end of the crouch phase or start of the take-off sequence in the hindlimb, except for the hip long axis rotation peak which occurs in the launch phase as the hindlimbs assume the flight pose.

A complication in the muscular reconstruction process are the prominent differences between crocodilian and avian estimations in the presence or absence of different muscles. The TM, HR, FDL.U, and TR.M are absent in birds, and the TR.C is variable between species but are all present in crocodilians; conversely the FCR is not present in crocodilians (*Dilkes, 1999*; *Bennett, 2003*; *Bennett, 2008*). The inclusion of these muscles results in changes to the muscle reconstruction and the resultant moment arms of the model with entirely avian based moment arms being slightly reduced compared to purely crocodilian or combined models (See Supplemental Information). While it is possible to determine the presence of

some muscles *via* muscle scars left on the fossils it is important for the inferences to be clear for the results of any modelling attempts.

While the ankle joint would be utilised at some stage in all the hypothesised take-offs, our model was not able to reconstruct this joint with precision due to the missing model bones in the reference specimen. Attempting muscle reconstruction without accurate models of the missing bones would result in substantial error from uncertainty in the origin and insertion points of the muscles (*Meilak et al., 2021a*). Other studies that have examined the moments produced around the ankle in crocodilians and birds (*Meilak et al., 2021a*; *Meilak et al., 2021b*; *Wiseman et al., 2021*). As both crocodilians and pterosaurs are plantigrade (*Mazin et al., 2003*; *Mazin & Pouech, 2020*), and lack the tibiotarsus seen in birds, a crocodilian ankle mechanics approach may be a closer approximation if the ankle muscles were to be estimated. These studies found the moments produced by crocodiles to peak at be around half the peak moment of the knee (*Wiseman et al., 2021*) while birds tended to peak at moment values equal or greater than the knee (*Meilak et al., 2021a*; *Meilak et al., 2021b*). If such results are applied to pterosaurs, it is unlikely for either of the bipedal take-off motions reach an equivalent amount of leverage as that available to the quadrupedal launch motion without some utilisation of the greater leverage available in the forelimb. If solely extensors are considered then the leverage available is similar between all three take-off motions with the burst take-off potentially overtaking the quadrupedal at the end of the sequence.

When proposing a quadrupedal take-off, *Habib (2008)* determined that the forelimbs of pterosaurs are stronger than the hindlimbs and as a result were likely able to withstand loads associated with quadrupedal launch. Our results further support this finding by determining that the muscle moment arms of the forelimb would be able to exert are also larger than the hindlimb, allowing pterosaurs to utilise the greater force resistance highlighted by *Habib (2008)*. Our own prior research into pterosaur range of motion found that ornithocheiraean pterosaurs could likely assume the poses required to quadrupedally take-off even when constrained by soft tissues (*Griffin et al., 2022*). The OpenSim model results indirectly lend support to the water take-off findings of *Pittman et al. (2022)* by highlighting the leverage possible in both the fore and hindlimbs which could be used to power water take-offs. These results are contrary to the findings presented by *Padian et al. (2021)* however that study focuses on a different pterosaur morphology, that of the giant azhdarchids. This previous study also highlighted the lack of moment arm analyses for the different launch hypotheses and raised concerns regarding the use of bats as models for quadrupedal take-offs due to the lack of hindlimb use by bats. This study addresses the moment arm concerns by determining that the moment arms of three different pterosaur take-off kinematics. Despite the bipedal take-off motions not having greater hindlimb moment arms or the quadrupedal take-off motions having consistently larger moment arms, the launch leverage available is greatest when utilising the quadrupedal take-off motion as a result of access to the larger moment arms available in the upper forelimb using any of the take-off motions (Fig. 4). Due to the lack of other recorded quadrupedally launching modern fliers from which to compare and derive timings, bat-based timing for the modelling of take-off in pterosaurs remains unavoidable though we have also expanded

the bat timing to incorporated hindlimb kinematics into the pterosaur take-off models. Future work may be able to refine the kinematics and better address this concern, potentially through iterative optimisation of kinematic approaches (*Bishop, Cuff & Hutchinson, 2021*).

When comparing hypothetical kinematic sequences, moment arms can only showcase the leverages available, they cannot inherently support one pose or kinematic sequence over the other as extant animals will use a variety of different poses with varying levels of moment arm support depending on the movements they are making *e.g.*, low walk *vs* high walk in crocodilians (*Wiseman et al., 2021*). Equally, a large moment arm does not inherently result in the creation of a larger force than shorter moment arms as an increased muscle mass can offset the effect of shorter moment arms. All three of the kinematic sequences the hypothetical pterosaur take-offs are based on are known to be capable of producing a successful take-off in extant animals (*Schutt Jr et al., 1997*; *Earls, 2000*) and are not assumed to be limited by any differences in the leverage produced by these motions. To get a better understanding of the overall moments generated by these kinematic sequences further work is also needed to apply muscular force estimations to the moment arms calculated for the pterosaur. The inclusion of muscular force estimations would also account for potential mismatches between the implied force available with greater leverage and the effects of muscle size. Additionally, the methods utilised in this study also need to be applied to the azhdarchid and non-pterodactyloid pterosaur morphologies to better facilitate comparison and develop a more complete understanding of pterosaur take-off.

## ACKNOWLEDGEMENTS

The authors would like to acknowledge the μ-VIS X-ray Imaging Centre at the University of Southampton for the provision of tomographic imaging facilities used in the original scanning of SMNK 1133.

### Funding

This work was supported by grants from the Bob Savage Memorial Fund managed by the University of Bristol, the Alan & Charlotte Welch fund managed by the Geological Society of London, Conselho Nacional de Desenvolvimento Científico e Tecnológico (CNPq) (grant #404352/2023-5 and #406902/2022-4) and São Paulo Research Foundation (FAPESP) (grant #2022/03099-7, #2019/10231-6, and 2023/11296-0). Emily J. Rayfield received the BBSRC grant BB/W00867X/1 for the APC. The funders had no role in study design, data collection and analysis, decision to publish, or preparation of the manuscript.

### Grant Disclosures

The following grant information was disclosed by the authors:
Geological Society of London, Conselho Nacional de Desenvolvimento Científico e Tecnológico (CNPq): 404352/2023-5 and 406902/2022-4.
São Paulo Research Foundation (FAPESP): #2022/03099-7, #2019/10231-6, and 2023/11296-0.

BBSRC: BB/W00867X/1.

## Competing Interests

Colin Palmer is an Academic Editor for PeerJ.

## Author Contributions

- Benjamin W. Griffin conceived and designed the experiments, performed the experiments, analyzed the data, prepared figures and/or tables, authored or reviewed drafts of the article, and approved the final draft.
- Elizabeth Martin-Silverstone conceived and designed the experiments, analyzed the data, authored or reviewed drafts of the article, and approved the final draft.
- Rodrigo V. Pêgas analyzed the data, authored or reviewed drafts of the article, and approved the final draft.
- Erik Anthony Meilak analyzed the data, authored or reviewed drafts of the article, and approved the final draft.
- Fabiana R. Costa analyzed the data, authored or reviewed drafts of the article, and approved the final draft.
- Colin Palmer conceived and designed the experiments, analyzed the data, authored or reviewed drafts of the article, and approved the final draft.
- Emily J. Rayfield conceived and designed the experiments, analyzed the data, authored or reviewed drafts of the article, and approved the final draft.

## Data Availability

The R scripts for randomly varying moment arm outputs are available in the Supplementary Files and Github: https://github.com/BWGriffin/PhD-Codes.

Tables detailing muscle attachment sites are also available in the Supplemental Files.

Full raw data along with the associated OpenSimm model (including the bone meshes and motion files) are available at the University of Bristol repository: Emily Rayfield, Elizabeth Martin, Benjamin Griffin (2024): Griffin_pterosaur_moment_arms NEW. https://doi.org/10.5523/bris.314kcutovxtty25yua4ptpjwk1.

## Supplemental Information

Supplemental information for this article can be found online at http://dx.doi.org/10.7717/peerj.17678#supplemental-information.

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
