# Peer review of "Modelling take-off moment arms in an ornithocheiraean pterosaur"

_PeerJ, doi:10.7717/peerj.17678_

## Round 0.1 · original submission · Major Revisions

I congratulate you on submitting a to-the-point, clearly written, and well-organized manuscript with to my knowledge the first comparative analysis of competing hypotheses and modelling of take-off moment arms in ornithocheiraean pterosaurs. I apologize for the delay in making my decision but I wanted to obtain 3 reviews and balancing their recommendations with my own and this proved more time-consuming than anticipated. Your study is also of broader relevance to other flying vertebrates. I therefore agree with the reviewers that the manuscript is of great importance and relevance to be published but some crucial points need to be addressed before publication:

1) Hypotheses and rationale: I appreciate, like reviewer 2, that the manuscript is concise and unnecessary difficult jargon is kept to the absolute minimum (but see also reviewer 3). I agree with reviewer 1 that the manuscript would benefit (e.g., would be easier to follow) from more clearly articulating the hypotheses and rationale behind your study (e.g., that muscle leverage limits takeoff ability; what actions and what joints might be limiting factors, why this particular group or specimen were chosen). As I am not a pterosaur paleobiologist some aspects were less straightforward to follow. Articulating those ideas more clearly in hypotheses would help with the greater reproducibility and broader understanding of your work. I feel that breaking down the aim of estimating leverage into specific hypotheses as suggested by reviewer 2 (e.g., summed HL extensor moment arms during the launch phase will be greatest in bipedal burst because [reference to experimental study]) would further advance this aspect of your study. I feel providing additional reconstructions or sketches would also contribute to this task and should at least be considered (see my suggestions in the annotated pdf). I agree with reviewer 1 that more clearly articulating the link between moments arms and launch ability would also benefit the discussion/interpretations of the results.


2) Summed moments arms: overall the methods are well documented, simulations well designed and limitations clearly stated. There are however some crucial choices and details which need additional detail to fully reproduce your analyses and follow your interpretations. Reviewer 1 raised the following points which need clarification: the decisions behind how many muscles to model, how many lines of action might influence the results of muscles as well as the reporting of maximum and/or mean moment arms. Reviewer 2 raised the following points which need clarification: definition of degrees of freedom in your plots of summed moment arms across forelimb and hindlimb joints, directions of flexion or adduction of the limbs, definition and illustration of the coordinate systems used in your study, correspondence between axes and degrees of freedom, how do movements in muscle moment arm correspond to movements in launch and changes in joint angle and kinematics as well as moment arm results for individual muscles.
3) Comparison to bird locomotion: Please address the suggestion of reviewer 3 to identify muscles as particularly large or important in pterosaurs and discuss the potential mismatches between high muscle moment arms but low muscle size.

4) Reorganization of discussion: I agree with reviewer 1 that starting out with a description of the motions involved in launch and which ones might be limited by muscle leverage as well as a figure or table comparing the MAs at these key points across the different launch strategies would benefit the manuscript.

5) Availability: I commend you (just like reviewers 2 and 3) for making available all data, models, and code. Reviewer 3 noted the zip folder with raw output data from OpenSim to be unavailable at the moment, please make sure all (raw) data, models, and code are available upon publication. I also agree with reviewer 3 that it also might make sense to do a final check of the comments to the code as they will be shared publicly.

6) Supporting references: Some statements might need rephrasing (compare reviewer 2) or additional support from published references (compare annotated pdf)


Please make sure to address these as well as all other points raised by the reviewers and myself (including annotated PDFs).

I look forward to receiving your revised manuscript.


Reviewer 1 ·

Basic reporting

The manuscript is clearly written and well organized, with sufficient background to understand the topic and the approach. The quality of the figures is excellent. It would be stronger if the hypotheses and their rationale were more clearly articulated. Specifically, the idea that muscle leverage limits takeoff ability should be justified by studies in extant animals. This sets up a problem for the discussion, because without an idea of what actions at what joints might be limiting factors it’s really hard to interpret the results. To improve this section, I would suggest that the aim of estimating leverage be broken into specific hypotheses, e.g., summed HL extensor moment arms during the launch phase will greatest in bipedal burst because [reference to experimental study].

Experimental design

The experiment is well designed. The research addresses a scientific question of great scientific and public interest, the flight ability of large bodied pterosaurs, from a new perspective (muscle leverage). Specifically, they aim to test whether different hypothesized launch strategies differ in terms of summed muscle moment arms. The methods are well described and documented, and their limitations are clearly stated. Using Monte Carlo simulations to estimate the effect of errors in moment arm calculation is a good idea. However, there is one additional limitation that should be addressed. The authors analyze summed moment arms, which are sensitive to the number of muscles modeled. Therefore, the decisions about how many muscles to model in the FL vs. HL and how many lines of action to use for a particular muscle will influence the results. Justification and possible effects of this methodological choice would be helpful. The authors might also consider reporting maximum and/or mean moment arms if there is a big difference.

Validity of the findings

The results and underlying data are clearly presented, but the interpretation is difficult because link between moment arms and launch ability is not fully articulated. The text states that the largest moment arms occur in the quadrupedal takeoff profile [265], but only a few of these actions would actually contribute to takeoff velocity. The authors acknowledge this to some extent (line 282). I suggest reorganizing the Discussion so that it starts out with a description of the motions involved in launch and which ones might be limited by muscle leverage (e.g., shoulder extension during the launch phase of quad launch). Also, I’d like to see a figure or table comparing the MAs at these key points across the different launch strategies and a discussion of these results. This discussion could include what muscles are mainly responsible for these differences and why, and whether there are any skeletal differences between pterosaurs, birds, and bats that might be relevant to the question of leverage. The idea of adding muscle force estimates in the future is intriguing [334-335].

Additional comments

Can you explain the statement about water leverage (319-320)?

·

Basic reporting

This manuscript is very well written - it is clear and concise. There is no obvious "fluff" or unnecessary use of difficult jargon. I am particularly impressed that the authors are sharing essentially everything: data, models, and code.

Experimental design

The research question is well defined, and the experimental design is perfect for the task. This is exactly the kind of approach that has been long needed to test the competing launch hypotheses for pterosaurs. By using a first-principles approach (moment arm comparisons), the authors have really broadened the scope of the paper beyond pterosaurs: it informs our understanding of animal launch and flight broadly, using pterosaurs as the exemplar. Very well crafted - and given the that the model and scripts are shared - quite easy to replicate (or even alter and use with a different taxon of interest).

Validity of the findings

Excellent transparency overall (see notes on data and model sharing above). The conclusions are straightforward and well stated - they are neither "overbaked" nor understated.

Additional comments

Overall, this paper is basically good to go in my opinion. I do have two very minor wording change suggestions for the introduction. These are not strictly necessary and could easily be taken care of in proofs if you all agree that they are worthwhile. I am, therefore, posting an official recommendation of "publish as-is" but including my little suggestions below for inclusion at your discretion.

“specific adaptations to achieve launch, thrust, and lift”
Given that thrust in flying animals is lift-based, I would recommend revising to “launch, thrust, and weight support (the latter two being lift-based)” or something similar.


“The most energy intensive part of powered flight is take-off from the ground as this requires the animal to generate enough velocity to overcome drag and for the wings to start to produce lift (Pennycuick, 1968; Rayner, 1988; Alexander, 1998). The take-off also requires the animal to get high enough into the air to start an unobstructed flapping cycle.”

This isn’t bad, but I think the wording here could use a slight bit of revision. “… generate enough velocity to overcome drag” is potentially a bit misleading. It’s not wrong, but it reads as a little imprecise (since the velocity overcomes drag indirectly - i.e., going fast enough that the wings produce enough lift that they can overcome drag). I would also point out that because take-off is energy intensive, it is size limiting (most readers will get this, but always good to just close the loop). It may also be better to talk about power here, instead of energy expenditure, since you’re looking at muscle moment arms. An example of a potential revision would be:

“The most power-intensive part of powered flight is take-off from the ground. This stage requires that the animal get high enough into the air to start an unobstructed flapping cycle. It also requires that the animal generate enough velocity such that the wings, when they do begin their first flapping cycle, produce enough lift to both overcome drag (e.g., thrust) and support the animal’s weight (Pennycuick, 1968; Rayner, 1988; Alexander, 1998). The height and velocity requirements both increase with larger size, making take-off size-limiting for flying animals.”

Reviewer 3 ·

Basic reporting

In terms of basic reporting, my one comment is that the paper seems quite short; the authors can afford to spend more time (and words) going over essential information and background. Otherwise, the paper satisfies the reporting requirements of the journal.

Experimental design

The research question is well defined and meaningful - how pterosaurs got airborne has been debated for many years within the paleontological community. This paper continues building on previous work by the same authors comparing across the different proposed launch styles and testing which was most biomechanically feasible. Therefore, this paper addresses a known knowledge gap in the field.

I do not fault the analyses the authors present in the paper, but I think more detail needs to be added before the paper is ready for publication. The authors present their results as a series of plots showing the summed moment arms for different degrees of freedom across the forelimb and hindlimb joints. However, at no point do the authors define what these degrees of freedom *are*. Which directions correspond to 'flexion' or 'adduction' of the limbs for example? Is one motion vertical and the other horizontal? Defining and illustrating the coordinate systems being used is a standard part of studies such as this. I see a figure in the supplemental information illustrating the joint coordinate systems, but it lacks sufficient annotation - which axes correspond to which degrees of freedom? Please add further annotations explaining which axes correspond to which directions to this supplementary figure, and move it into the main text of the paper.

Related to my point about joint axes, the authors illustrate each launch cycle to show the movements involved. However, they do not report the actual joint angles or how these change throughout the different phases of launch for each launch style. This makes it unclear how the observed changes in muscle moment arms correspond to the movements involved in launch. Please add plots of the launch kinematics to the appropriate Figures in the Results section so that readers of the paper can properly relate the changes in muscle moment arms to changes in joint angle and kinematics.

The authors also only discuss the summed results of their OpenSim models. There is no discussion of the moment arm results for individual muscles. Base on studies of bird locomotion, are there particular muscles that have been identified as particularly large or potentially important in pterosaur locomotion? If so, then it would make sense to highlight these individual muscles. This might assuage any concern over mismatches between high muscle moment arms but low muscle size (and so ultimately low moment generating capacity).

Validity of the findings

The conclusions of the paper are well stated, but I think more data needs to be presented with the paper in order to fully assess how well they are supported. Not all underlying data were made available for review with the manuscript (I cannot find the Zip folder MMA containing the raw output data from OpenSim), but the authors promise to make al data necessary for replication available on publication.

As a note to the authors, please check the comments on your code; this is being made freely available, other people will read it!

---

## Round 0.2 · Minor Revisions

Thanks you for revising your manuscript which makes the research easier to follow and reproduce and of even greater value to the community. I would love to see this interesting approach published, but some crucial points remains to be addressed before publication:

Discrepancies between hypotheses stated in the introduction versus the discussion and abstract: the hypotheses are now more clearly stated in the introduction which makes the manuscript easier to follow, but they are not consistent with what is reported in the discussion and abstract (compare reviewer 1). In the introduction you focus on comparing hindlimb versus forelimb leverage while in the abstract and discussion you focus on comparing total leverage (forelimb and hindlimb). Please homogenize and justify the use of the total arm movement (compare reviewer 1)

Justify that muscle leverage limits takeoff by studies in extant animals: I have understanding for that fact that you do not want perform additional analyses if these are already planned for a future study. However, I agree with reviewer 1 that at minimum references from the literature should be cited which would support such an idea/rationale.

Clarify use of maximum, mean and summed moments arms across muscles versus range of motion: Please consider addressing the suggestion by reviewer 1 to also report maximum and mean moment arms as well as summed moment arms across muscles

Description of the motions involved in launch and limited by muscle leverage: I agree with reviewer 1 that it would be crucial to describe the motions involved in launch and which ones might be limited by muscle leverage. Also adding a figure or table comparing the moment arms at these key points across the different launch strategies and a discussion of these results would be critical. I agree with reviewer 1 that because the motions contributing to launch and moment arms are not considered separately, the discussion is oversimplified and some interpretations likely not justified. Please, discuss the complexity of your approach and potential pitfalls (or how one can deal with them) in greater detail and make sure your conclusions are supported by your results. I agree with reviewer 1 that the idea that differences in summed moment arms across the entire limb can support one pose over another requires further clarification and justification.

Absolute or relative values: you describe trends in your results but do not provide values or comparisons in text (e.g., effect size). How much do values changes within particular movements? How do values compare to (total) moment arms and how they (relatively) compare between different movements. If absolute values are not biologically meaningful for comparisons, this should at least be stated/explained and relative values/comparisons (e.g., percentage) should be used. It is hard to assess effect size and compare between movements without such values.

References: The references concerning previous work in modern flying vertebrates (e.g., birds as well as bats) or other relevant studies for your approach or particular statements remain limited. Please make sure at least one and ideally at least two references are cited for crucial statements.

Please make sure these as well as all others points raised including those in annotated pdfs are addressed. I look forward to receiving your revised manuscript and seeing this valuable work published.

Reviewer 1 ·

Basic reporting

The revised version is improved by having the hypotheses clearly stated at the end of the introduction.
However, these hypotheses do not seem consistent with what is reported in the discussion and abstract. In the introduction, the emphasis is on comparing hindlimb vs. forelimb leverage during the launch phase in different take-off poses: "we hypothesise that the leverage of the hindlimbs during the launch phase would be greatest in the bipedal burst followed by the countermotion motions ... while the leverage of the forelimbs will be greater in the quadrupedal launch phase than either of the bipedal take-offs." Yet, in the abstract and discussion the emphasis is on comparing TOTAL leverage (forelimb plus hindlimb): "we found the muscles utilised in the quadrupedal take-off have the largest moment arms ... " (lines 20-22) and "Overall, when all of the moment arms are considered, the quadrupedal take-off summed moment arms are greater in magnitude than moment arms associated with the other two take-offs scenarios." The "total moment arm" comparison may be valid, but because it's not formally proposed there's no explanation or rationale.

Also, both Reviewer 3 and I requested that the idea that muscle leverage limits takeoff ability should be justified by studies in extant animals. The authors declined to do so because they are planning to make this the subject of a future study. I should clarify that I was not asking for new experiments, just references from the literature supporting this idea. I think this is necessary background for the current manuscript.

Experimental design

The experimental design is sound. The revision satisfactorily addresses my question about why certain muscles were included and how muscles with most lines of action were handled. To clarify, when I suggested reporting maximum and mean moment arms as well as summed moment arms I meant across muscles, not across range of motion. However, it's just a suggestion and I don't think it's absolutely necessary.

Validity of the findings

The discussion has been reorganized with the hypotheses in the beginning, and I think this is effective. I had also suggested inclusion of a description of the motions involved in launch and which ones might be limited by muscle leverage, which was not done. In addition, I suggested including a figure or table comparing the MAs at these key points across the different launch strategies and a discussion of these results. The authors declined to do so because the key points are different in each take-off motion. Because the motions contributing to launch and the moment arms associated with these motions are not considered separately, I think that the discussion currently is too simplistic and some interpretations are unjustified. For example, shoulder and elbow extension MA during launch, which probably contribute greatly to takeoff power, are actually smaller in the quad launch than bipedal or burst. Yet, the discussion states that "the quadrupedal take-off summed moment arms are greater in magnitude than moment arms associated with the other two take-offs scenarios." (lines 335-337), ignoring the difference between actions that are likely to be important for launch and ones that are not (e.g., shoulder flexion and lateral rotation). Therefore, while the results are valuable, I am not convinced that they support the conclusions.

Additional comments

I appreciate the changes the authors have made in response to my and the other reviewers' suggestions. However, I feel that in some areas these changes did not go far enough. In particular, the idea that differences in summed moment arms across the entire limb can support one pose over another requires justification.

Reviewer 3 ·

Basic reporting

The manuscript is well written. I appreciate the expanded Introduction and Discussion. I am much happier now with the overall layout of the paper.

Experimental design

I would like to thank the authors for adding additional methodological details, both in the text and figures. I believe my concerns from the last round have been sufficiently addressed.

Validity of the findings

With the additional reporting, I am confident in the conclusions drawn by the authors. I think the paper is now ready for publication.

---

## Round 0.3 · Minor Revisions

The revisions particularly homogenizing the hypotheses, adding additional modern references and adding the mean, maximum and summed moment arms make your study easier to follow and reproduce. I agree with reviewer 1 that there are still some aspects which need to be addressed before publication:

1) Add non-kinematic references: Please provide references that discuss muscles or leverage on modern vertebrates that justify that muscle leverage limits takeoff (compare reviewer 1)
2) Justify and consider alternatives to adding moment arms: I agree with reviewer 1 that not all moment arms described as “launch applicable” contribute to launch power and that different considerations may affect your conclusions. . The most critical issue that needs to be addressed as stated by reviewer 1 is: if only extensor or "stance phase" actions were considered to be part propulsion, which common practice in biomechanical modeling, the conclusion would be different. I would like to at least see these aspects acknowledged and ideally analysed/discussed in greater detail in your paper.

Please address these as well as other points raised. I look forward to receiving your revised manuscript.

Reviewer 1 ·

Basic reporting

The hypotheses are now clear and consistent throughout the manuscript.

Additional references to studies on extant animals have been added (lines 60-108). However, they do not address the comment "Justify that muscle leverage limits takeoff by studies in extant animals" because they are kinematic studies only and do not discuss muscles or leverage. Also, there are still no references to work on modern flying vertebrates in the discussion.

The decision to report mean, maximum, and summed moment arms in SI seems reasonable. I'm very glad to see values reported in the text - it makes the results much clearer.

Specific comments:
272 - should this be "external rotators" not "extensors"?
318 - I think this should say 0.056, not 0.56
355 - Please clarify whether the second 0.144m refers to extension or external rotation

Experimental design

no comment

Validity of the findings

The reason I asked for a kinematic description is not to "support one pose over another," but because many of the moment arms described as “launch applicable” (e.g., in the abstract) almost certainly do not contribute to launch power. For example, the highest moment arms were in elbow flexion, which would not contribute to launch because the elbows are extended throughout that phase. If you compare only extension, which probably is the most relevant for generating propulsive forces (see, e.g., Hutchinson 2004's paper on running in extant birds), the forelimb and hindlimb moment arms are very similar. Thus, the decision to add all moment arms together whether or not they relate to the kinematics of launch changes the conclusions of the paper. That is why this decision needs to be justified and alternatives considered.

At a minimum, it could say something like, “however, combined shoulder and elbow extensor moment arms in the forelimb during quadrupedal launch were similar to combined hip and knee extensor moment arms in the hindlimb.”

(both FL and HL have additional joints that could be involved in takeoff, as noted in the ms, but we can’t compare them since the ankle is missing.)

Hutchinson, J.R. “Biomechanical Modeling and Sensitivity Analysis of Bipedal Running Ability. I. Extant Taxa.” Journal of Morphology 262, no. 1 (2004): 421–40.

Additional comments

The revisions have improved the manuscript greatly, especially the values reported in the text. The most important thing that still needs to be addressed is: if only extensor or "stance phase" actions were considered to be part propulsion, which common practice in biomechanical modeling, the conclusion would be different.

---

## Round 0.4 · accepted · Accept

Thank you for addressing these final suggestions. The implemented changes make the manuscript even more consistent and easier to follow. The added references provide additional context and make it easier to follow the rationale behind your approach. I look forward to see this interested work published.

Reviewer 1 ·

Basic reporting

no comment

Experimental design

no comment

Validity of the findings

no comment

Additional comments

I appreciate the addition of the "extension only" takeoff to SI as this will make it easier to compare with other modeling studies.
I understand that the references I would like to see, that relate moment arms to take-off in extant birds, may not exist in the experimental literature. Hopefully someone will be inspired to fill this knowledge gap!